# An HIV-1/HIV-2 Chimeric Envelope Glycoprotein Generates Binding and Neutralising Antibodies against HIV-1 and HIV-2 Isolates

**DOI:** 10.3390/ijms24109077

**Published:** 2023-05-22

**Authors:** Nuno Taveira, Inês Figueiredo, Rita Calado, Francisco Martin, Inês Bártolo, José M. Marcelino, Pedro Borrego, Fernando Cardoso, Helena Barroso

**Affiliations:** 1Centro de Investigação Interdisciplinar Egas Moniz (CiiEM), Egas Moniz School of Health and Science, 2829-511 Caparica, Portugal; 2Research Institute for Medicines (iMed.ULisboa), Faculty of Pharmacy, University of Lisbon, 1649-003 Lisboa, Portugal; 3Centre for Public Administration and Public Policies, Institute of Social and Political Sciences, Universidade de Lisboa, 1300-663 Lisbon, Portugal; 4Unidade de Microbiologia Médica, Saúde Global e Medicina Tropical, Instituto de Higiene e Medicina Tropical, Universidade NOVA de Lisboa, 1099-085 Lisbon, Portugal

**Keywords:** HIV vaccine, chimeric envelope glycoproteins, vaccinia virus, neutralizing antibodies, neutralizing epitopes

## Abstract

The development of immunogens that elicit broadly reactive neutralising antibodies (bNAbs) is the highest priority for an HIV vaccine. We have shown that a prime-boost vaccination strategy with vaccinia virus expressing the envelope glycoprotein gp120 of HIV-2 and a polypeptide comprising the envelope regions C2, V3 and C3 elicits bNAbs against HIV-2. We hypothesised that a chimeric envelope gp120 containing the C2, V3 and C3 regions of HIV-2 and the remaining parts of HIV-1 would elicit a neutralising response against HIV-1 and HIV-2. This chimeric envelope was synthesised and expressed in vaccinia virus. Balb/c mice primed with the recombinant vaccinia virus and boosted with an HIV-2 C2V3C3 polypeptide or monomeric gp120 from a CRF01_AG HIV-1 isolate produced antibodies that neutralised >60% (serum dilution 1:40) of a primary HIV-2 isolate. Four out of nine mice also produced antibodies that neutralised at least one HIV-1 isolate. Neutralising epitope specificity was assessed using a panel of HIV-1 TRO.11 pseudoviruses with key neutralising epitopes disrupted by alanine substitution (N160A in V2; N278A in the CD4 binding site region; N332A in the high mannose patch). The neutralisation of the mutant pseudoviruses was reduced or abolished in one mouse, suggesting that neutralising antibodies target the three major neutralising epitopes in the HIV-1 envelope gp120. These results provide proof of concept for chimeric HIV-1/HIV-2 envelope glycoproteins as vaccine immunogens that can direct the antibody response against neutralising epitopes in the HIV-1 and HIV-2 surface glycoproteins.

## 1. Introduction

Acquired immunodeficiency syndrome (AIDS) is caused by human immunodeficiency virus type 1 (HIV-1) and type 2 (HIV-2). While HIV-1 is prevalent worldwide, HIV-2 is largely confined to West Africa and countries with socio-economic ties to the region, such as Portugal and France [1]. In 2021, there were 38.4 million people living with HIV of which an estimated 1–2 million people worldwide had HIV-2 [2,3]. Although new HIV infections have fallen by 52% since the peak in 1997, an estimated 1.5 million people became newly infected in 2021 [3,4] and this number is expected to increase due to the COVID-19 pandemic [5]. A vaccine is the best hope for preventing HIV transmission and ending the AIDS pandemic. However, despite more than 30 years of research, there’s still no effective HIV vaccine. The high diversity and rapid evolution of the virus are major hurdles to the development of a prophylactic HIV vaccine (reviewed in [6]). An effective vaccine must induce antibodies capable of neutralizing tier 2 and tier 3 viruses of all clades. These broadly neutralizing antibodies (bNAbs) are produced in 5% to 30% of adult individuals with HIV-1 after several years of infection [7,8,9,10,11,12]. Vaccine strategies to cope with the high diversity of HIV-1 have used multivalent mixtures of immunogens and ancestral, consensus or mosaic proteins that are computationally designed to elicit immune responses with improved cross-reactive breadth [13,14,15,16,17,18,19,20,21,22,23]. Unfortunately, to date, antibodies elicited by candidate immunogens and vaccines have shown a limited ability to neutralize heterologous primary HIV-1 strains [19,20,22,24,25,26,27,28,29,30,31,32].

bNAbs target seven conserved epitopes in the HIV-1 envelope: the CD4-binding site (CD4bs), the CD4-induced site, the V2 apex, the N332-dependent high mannose patch epitope in V3, the 2G12 N-glycan epitope, the gp41 MPER and the gp41/gp120 interface which contains the fusion peptide [6,33,34,35]. The most common, potent and broad bNabs target the CD4bs, the V2 apex, and the V3-glycan patch epitopes in the HIV-1 envelope. CD4bs antibodies bind the CD4 binding site of gp120 and belong to two major families: those that mainly rely on the heavy chain complementary determining region 2 (HCDR2) to interact with HIV-1 gp120 (CD4 mimic bNAbs, e.g., B12, VRC01 class and 8ANC131/CH235 class) and those that instead use HCDR3 to contact gp120 (HCDR3-binding bNAbs, e.g., CH103) (reviewed in [23,36]). The epitope of V3-glycan targeting bNAbs is located at the base of the V3 region between N-linked glycans at positions 301 and 332. Their long HCDR3 loops are needed to reach the peptide backbone beyond the dense glycan patch defined by the GDIR sequence (amino acids 324–327). Examples of these bNAbs are PGT121, PGT128 and 447-52D. bNAbs targeting the V2 apex typically have extra-long HCD3 loops and sulfated tyrosine motifs required to contact the peptide epitope, a lysine-rich strand around positions 168–171, under the N-linked glycan at position 160 [23,36]. Examples of such bNAbs are PG9, PG16 and VRC26.25. Directing the immune system to elicit such bNAbs remains a major challenge due to the extremely complex antibody maturation pathways and high levels of somatic hypermutation required by most HIV-1-specific antibodies to acquire neutralizing breadth [6,23,37,38]. 

Like HIV-1, HIV-2 is a highly variable virus and consists of nine groups called A to I, of which group A is by far the most prevalent (reviewed in [39]). HIV-1 and HIV-2 show remarkable differences in the course of the infection (reviewed in [40,41]). The most striking differences include the low to undetectable plasma viral load in 80% of untreated individuals with HIV-2, the much lower rate of activation and depletion of CD4+ T cells per year in HIV-2 infection and the production of broadly neutralizing antibodies in most (90–100%) adult individuals with HIV-2. Consequently, compared to HIV-1, HIV-2 infection is characterized by a much longer asymptomatic phase, slower clinical progression and lower transmission rates. Although the course of HIV-2 infection is longer than that of HIV-1, without effective antiretroviral therapy, a significant proportion of infected individuals will progress to AIDS and die [42,43]. 

As bNAbs are considered to be the best correlate of protection against HIV infection, the development of envelope immunogens that elicit bNAbs is currently the main priority for the HIV vaccine field [6,44,45,46]. As mentioned above, to date, no HIV-1 vaccine immunogen has been able to consistently elicit bNAbs in relevant animal models and in humans [23]. In contrast, we have shown that the surface envelope glycoprotein and a polypeptide comprising the C2, V3 and C3 of HIV-2 strain ALI induce the production of potent V3-directed neutralizing antibodies in mice against several primary group A HIV-2 isolates [47].

In this work, we hypothesized that a chimeric envelope containing the V3 region of HIV-2 and the remaining parts of HIV-1 would elicit a bNAb response against both HIV-1 and HIV-2. This chimeric envelope was synthesized and expressed in vaccinia virus. Mice primed with this recombinant vaccinia virus and boosted with an HIV-2 C2V3C3 polypeptide or HIV-1 recombinant soluble HIV-1 gp120 from a CRF01_AG isolate (gp120AG) [22] generated HIV-1 and HIV-2 neutralizing antibodies which in the case of HIV-1 targeted the three major neutralizing epitopes in the envelope gp120.

## 2. Results

### 2.1. Expression and Antigenicity of HIV1/HIV2 Chimeric Envelope

Recombinant vaccinia virus expressing the chimeric HIV-1/HIV-2 envelope was obtained with infectious titres in the range of 10^12^–10^14^ PFU/mL. To characterize the expression of the chimeric Env, HeLa-CCL2 cells were infected with recombinant vaccinia (5 PFU/cell), and cell lysates were obtained and analysed by Western blot and ELISA assays. Western blot analysis with sera from individuals with HIV-1 and HIV-2 showed that the chimeric HIV-1/HIV-2 envelope gp120 was produced at high levels and reacted strongly with antibodies from individuals with HIV-1 and HIV-2 (Figure 1).

The binding reactivity of the soluble chimeric Env gp120 was analysed in ELISA assays against sera of individuals with HIV-1 as a positive control and against human bNAbs directed against CD4bs, the V1-V2 region, V3 and N-linked glycans in the C2, C3, V4 and C4 regions in gp120 and gp41 MPER and compared with the binding of two reference HIV-1 envelope glycoproteins. All nine gp120-specific monoclonal antibodies bound to the consensus gp120 (M Con. S D-11), whereas six of the nine gp120-specific antibodies and the gp41-specific antibody 2F5 bound to SF162 gp140 (Table 1). Antibodies B12 and 447-52D bound to the chimeric gp120, indicating the preservation and exposure of the CD4 binding site and V3 loop epitope structures recognized by these antibodies. The lower and narrower monoclonal binding reactivity of the chimeric gp120 compared to the reference gp120 and gp140 envelope glycoproteins suggests a different conformation and neutralizing epitope exposure.

### 2.2. Chimeric Env Induces HIV-1 and HIV2 Binding and Neutralising Antibodies in Mice

Balb/c mice were inoculated with recombinant vaccinia virus expressing the chimeric Env and boosted four times with soluble gp120_AG or HIV-1 ALI C2V3C3 polypeptide (Figure 2). The mice produced increasing levels of antibodies that bound to the autologous antigens as assessed by ELISA (Figure 3). Mice boosted with the HIV-2 C2V3C3 polypeptide produced high-binding antibody responses to the cognate polypeptide (Figure 3A) and did not produce antibodies to gp120_AG (Figure 3C). In contrast, mice boosted with gp120_AG produced 2.2-fold higher binding antibody responses to the HIV-2 C2V3C3 polypeptide than to the cognate antigen after boost 4 (Figure 3, compare panels B and D). After the final booster immunisation, all mice produced antibodies that neutralised ≥60% (serum dilution 1:40) of the primary HIV-2 isolate HCC6.03, a CCR5-using virus (Figure 4; Appendix A) [44]. Two mice boosted with C2V3C3 polypeptide (group II) or gp120_AG (group III) also produced antibodies that neutralized the HIV-1 strain SG3.1. Finally, two mice boosted with gp120_AG (group III) produced antibodies that neutralized at least one tier 2 HIV-1 isolate. Of these, mouse 5 was able to produce antibodies that neutralized 97% of the HIV-2 isolate and ≥48% of the three HIV-1 isolates tested.

### 2.3. Epitope Mapping Analyses

HIV-1 neutralizing epitope specificities were assessed using a panel of HIV-1 TRO.11 mutant pseudoviruses whose neutralizing epitopes in V2, CD4 binding site or V3 high mannose patch were disrupted by the alanine substitution of selected amino acids defining the epitopes, namely, N160A in V2, N278A in the CD4 binding site and N332A in the V3 high mannose patch. In this assay, a decrease in the neutralizing potency of the mutants relative to wild-type TRO.11 indicates that the neutralizing antibodies target the epitopes disrupted by the alanine mutation [48]. Group III mouse 5 was unable to neutralize the TRO.11 mutants, indicating that its neutralizing antibodies were directed against the three epitopes (Figure 5). In contrast, group III mouse 4 still retained some neutralizing activity (>30%) against all mutants, suggesting that its neutralizing response was directed against other neutralising epitopes in gp120.

## 3. Discussion

The development of an Env-based immunogen capable of eliciting broadly neutralizing antibodies is a priority in the roadmap to produce an effective HIV vaccine and end the AIDS epidemic by 2030 [6]. This study investigated the antigenicity and immunogenicity of a novel chimeric envelope gp120 comprising the C2, V3 and C3 regions of HIV-2 and the remaining parts of the HIV-1 envelope. This construct was designed to allow the exposure of broadly reactive neutralizing epitopes in the envelope gp120 of HIV-1 (V2 apex and the CD4bs) and HIV-2 (V3 and the CD4bs) and to promote the elicitation of neutralizing antibodies against both types of virus. Chimeric or mosaic constructs containing T- and/or B-cell epitopes of Gag, Pol, Vif, Nef and Env designed to elicit HIV-specific antibodies and CD4- and CD8- T-cell responses have been used in several HIV-1 vaccine clinical trials (reviewed in [6]). These immunogens elicited Env-specific and V1V2-directed binding antibodies and antigen-specific T-cell responses, but no tier 2 neutralising antibodies were reported. HIV-1/HIV-2 chimeric envelope glycoproteins have not previously been used in vaccine trials. In Western blot, the chimeric gp120 bound to polyclonal serum antibodies from HIV-1- and HIV-2-infected individuals, confirming the presence of antigenic epitopes from both viruses. Mice primed with a vaccinia virus expressing the chimeric envelope and boosted with envelope gp120 from an HIV-1 CRF01_AG isolate [22] or a C2V3C3-polypeptide from HIV-2ALI [47] produced Env-specific binding and neutralizing antibodies against heterologous HIV-2 and tier 1 HIV-1 isolates. Two animals boosted with gp120AG also produced neutralizing antibodies against heterologous tier 2 HIV-1 isolates. Boosting with gp120AG produced broader and more potent neutralizing antibodies than boosting with the C2V3C3 polypeptide. The production of potent HIV-2 neutralizing antibodies by all mice irrespective of boosting modality confirms that the gp120 chimera adequately exposes the C2V3C3-based neutralizing epitopes of HIV-2 [47,49,50,51].

To investigate the exposure of the major neutralizing epitopes of HIV-1, we tested the binding reactivity of the chimeric gp120 to a panel of broadly neutralizing monoclonal antibodies targeting different conformational and N-linked epitopes. Previous studies have shown that the affinity of most bNAbs for monomeric envelope gp120/gp140 or core constructs of gp120 is variable and may not be related to their neutralizing efficacy [52,53]. In fact, with the exception of B12 and 447-52B, most monoclonal antibodies did not bind to chimeric gp120 in ELISA assays. The amino acids defining the B12 epitope are scattered in C1 (K97, L125), V2 (D167, D185, T202), C2 (N276), C3 (S365, P369), C4 (V430, T455, G458) and C5 (G473) [53,54,55]. Except for the substitutions E for P at position 369 in C3 and T for V at position 430 in C4, all amino acids are conserved in our chimeric construct, even those derived from HIV-2 in C2 (N276) and C3 (S365 and P369) (Appendix A). This high level of amino acid conservation in the context of a similar HIV-1 and HIV-2 gp120 structure [56] likely contributes to the binding of B12 to our chimeric gp120. While binding to B12 confirms that the chimeric gp120 exposes a CD4bs epitope, the lack of binding to the other antibodies that mimic the CD4 binding site suggests an altered CD4 binding site and impaired binding to the CD4 receptor in the cell surface. This is an advantageous feature for an Env immunogen as binding to cellular CD4 can lead to sequestration by CD4+ T cells and the occlusion of the CD4bs neutralizing epitopes [17]. The binding of 447-52B to the chimeric gp120 was unexpected because this antibody recognizes the conserved tip of the HIV-1 V3 loop in a β-turn conformation and preferentially binds and neutralizes HIV-1 isolates with the GPGR motif in the V3 crown [57,58,59]. Our chimera contains the HIV-2ALI V3 loop, the sequence (Appendix A) and structure of which, as determined by homology modelling [47], differs significantly from that of the 447-52B epitope. In addition, previous studies have shown that 447-52 is unable to neutralize two different strains of HIV-2, UC1 and 7312A [60]. Further studies are needed to determine whether a 447-52B-like neutralizing epitope is formed in our chimeric gp120. 

Epitope mapping experiments with the subtype B strain TRO.11 showed that the neutralizing antibodies elicited in mouse 5 target broadly reactive epitopes in V2, the CD4 binding site and the high mannose patch centred on N332 at the base of the V3 loop. The presence of CD4bs antibodies was not surprising given the conservation of the CD4 binding site amino acids in the chimeric gp120 (see above). The elicitation of neutralizing antibodies against the conserved N160/V2 epitope suggests that our gp120 chimera exposes this broadly reactive epitope in a way that is not captured by the PG16 monoclonal antibody but allows the expansion of B lymphocytes expressing germline precursors of this class of bNAbs [61]. The production of neutralizing antibodies targeting the N332 glycan-dependent supersite in V3 was surprising given that the asparagine residues supporting the major N332 glycan and other N-glycans that can contribute to antibody interaction, such as N295 in C2, N301 in V3 and N339 in C3, are not present in our construct and that the homologous region in the HIV-2 envelope lacks N-linked glycosylation sites [51,56,62]. However, other N-glycans around V3 and underlying peptides that come together in the tertiary structure of HIV-1 gp120 and may interact with antibodies of this class, namely, N133 in C1, N137 and N156 in V1, N160 in V2 and N385 and N392 in V4, are conserved in our construct [36,63,64]. This suggests that some neutralizing antibodies elicited in our animals interact with N-glycans and underlying peptides outside C2, V3 and C3, whose exposure in TRO.11 is significantly affected by the alanine substitution of N332. An example of this type of V3-glycan-directed bNAbs is PGT121, which also targets glycans at positions 137 and 156 in V1 and 160 in V2 [65]. 

There are several limitations to this study. First, the number of animals in each group was small. Second, the neutralizing antibody response in mice was tested against a limited number of HIV isolates. Third, the epitope mapping was focused only on three broadly neutralizing epitopes in HIV-1 gp120. The small amounts of serum obtained from these animals prevented the testing of neutralizing activity against a broader panel of isolates and more detailed epitope mapping studies. Further studies in larger animal models are required to better characterize the immunogenic potential of our gp120 chimera. Monoclonal antibody isolation is required to confirm the neutralizing epitope specificities found in the best-responding animal. 

## 4. Materials and Methods

### 4.1. Cells, Plasmids, Viruses, and Antibodies

Rat2 (TK) cells were obtained from the American Type Culture Collection (ATCC) (Rockville, MD, USA). HeLa cells (ATCC^®^CCL-2™) were obtained from the American Type Culture Collection. TZM-bl cells were provided by the AIDS Research and Reference Reagent Program (ARRRP), National Institutes of Health. HeLa, Rat2 and TZM-bl cells were cultured in complete growth medium consisting of Dulbecco’s minimal essential medium (DMEM) supplemented with 10% *v/v* fetal bovine serum (FBS), 100 U/mL penicillin-streptomycin (Gibco/Invitrogen, Waltham, MA, USA), 1 mM sodium pyruvate (Gibco/Invitrogen, Waltham, MA, USA), 1 mM L-glutamine (Gibco/Invitrogen, Waltham, MA, USA) and 1 mM non-essential amino acids (Gibco/Invitrogen, Waltham, MA, USA). All cell cultures were maintained at 37 °C in 5% CO_2_. 

The following were also obtained from the ARRRP: Western Reserve strain of vaccinia virus (VV_WR_), recombinant vaccinia virus vPE16 expressing the envelope gp120 of HTLV-IIIB clone BH8, a panel of global tier 2 HIV-1 Env clones (cat#12670) designed to assess neutralisation responses, HuMAbs PG9 (anti-V1V2), PG16 (anti-V2), 447-52D (anti-V3), 2G12 (anti-N-linked glycans in C2, C3, V4, C4), VRC01 (anti-CD4bs), VRC03 (anti-CD4bs), 3BNC117 (anti-CD4bs), HJ16 (anti-CD4bs), b12 (anti-CD4bs) and 2F5 (anti-MPER), and recombinant proteins M. CON.SD11 gp120 and SF162 gp140 trimer.

### 4.2. Production and Expression of Recombinant Vaccinia Viruses 

A codon-optimized chimeric envelope gene containing gp120 region of HIV-192TH023 (CRF01_AE, GenBank accession number: KU562843.1) with the C2V3C3 region of HIV-2ALI (group A, GenBank accession number: L25445.1) [66] and gp41 of HIV-1Ri112W2R074B10 (subtype B, GenBank accession number: JN205576) was produced by gene synthesis at NZYTech, Lisbon, Portugal (Figure 6; Appendix A). 

The chimeric env gene was subcloned into the BamHI and HindIII sites of the vaccinia virus insertion vector pMJ602. HeLa cells were transfected with recombinant pMJ602 plasmid and infected with 0.5 PFU of vaccinia virus strain VV_WR_, and recombinant vaccinia virus resistant to 5-bromodeoxyuridine and expressing β-galactosidase was selected in Rat2 cells and then propagated in HeLa cells [22,47,67]. The recombinant virus was titrated in Rat2 cells according to the method of Reed and Muench [68]. 

### 4.3. Characterization of Chimeric Envelope Glycoproteins

HeLa cells were infected with 5 PFU of recombinant vaccinia virus per cell and incubated for 1 h. After 24 h of infection, the cells were washed with cold phosphate-buffered saline (PBS) and lysed with RIPA+DOC buffer (0.15 M NaCl, 0.05 M Tris-HCl, 1% Triton X-100, 1% DOC, 0.1% SDS). The lysates were centrifuged at 35,000 rpm for 60 min at 4 °C and the supernatant containing the proteins was collected. The expression and antigenic reactivity of envelope glycoproteins were analysed by Western blot (WB) and ELISA. For the WB analysis, proteins separated by 7.5% SDS-PAGE were transferred to 0.45 µM nitrocellulose membranes (Bio-Rad) treated with blocking buffer (1X TBST with 4% w/v nonfat dry milk) and incubated with serum from individuals with HIV-1 and HIV-2 diluted (1:200) in primary antibody buffer (1× TBST with 4% w/v nonfat dry milk and 5% goat serum). The membrane was then washed with TBST 0.25% and incubated with goat anti-human IgG alkaline phosphatase antibody (Sigma, St. Louis, MO, USA) and/or goat anti-human IgG peroxidase antibody (Sigma). The colourimetric detection of proteins was performed using the AP Conjugate Substrate Kit (Bio-Rad, Hercules, CA, USA), and chemiluminescent detection was performed using Pierce™ ECL Western Blotting Substrate (ThermoFisher Scientific, Waltham, MA, USA). Envelope gp120 from HTLV-IIIB clone BH8 expressed by recombinant vaccinia virus VV_PE16_ was used as an HIV-1 positive control and recombinant vaccinia viruses expressing the full-length HIV-2ALI envelope (VV_ALI_), and a truncated form of gp125 (M2) were used as HIV-2 positive controls [47].

The method of Rose et al. [69] was used to obtain the soluble chimeric gp120. HeLa cells were infected with 5 PFU of recombinant vaccinia virus per cell and incubated for 3 h. The medium containing the infecting virus was replaced with DMEM supplemented with 2.5% FBS 3 h after infection. After 24 h of infection, medium containing released gp120 was collected, clarified by centrifugation at 3000× *g* for 10 min and filtered through a 0.2 µM pore size filter to remove the vaccinia virus.

The binding specificity of the chimeric gp120 was analysed in ELISA assays against ten broadly neutralizing monoclonal antibodies at a final concentration of 10 µg/mL: 2G12, b12, 447-52D, HJ16, PG9, PG16, 3BNC117, VRC01, VRC03 and 2F5. Immuno MaxiSorp 96-well microplates (Nunc) were coated with the chimeric env (~2.3 to 4 µg per well) and reference SF162 gp140 (2 µg/mL) and M.CON.S D11 gp120 (2 µg/mL) glycoproteins in 0.05M bicarbonate buffer. HIV-1 serum sample at a final dilution of 1:200 was used as a positive control and 2F5 antibody (anti-gp41) as a negative control. The cut-off value of the assay with the chimeric and M.CON.S D11 gp120s was calculated as the mean OD value of the 2F5 reactivity plus 3 times the standard deviation (SD). The cut-off value of the assay with HIV-positive serum and SF162 gp140 (which reacts with 2F5) was calculated as the mean OD value of HIV seronegative samples plus 3 times the SD.

### 4.4. Balb/c Mice Immunizations

Three groups (I, II and III) of 6-week-old female Balb/c mice were immunised intraperitoneally with 10^7^ PFU of wild-type vaccinia virus WR (VV_WR_) or vaccinia virus expressing the chimeric Env (VVChimeric ENV) in 100 µL phosphate-buffered saline (PBS) (priming) and on days 15, 30, 45 and 60 with 10 µg of purified recombinant HIV-2ALI C2V3C3 polypeptide or soluble gp120 from a clade CRF01_AG isolate (gp120AG) emulsified in incomplete Freud adjuvant (Figure 2). The production and purification of these booster proteins were previously described by Calado et al. [22]. Group I (two mice) was the control group and received VV_WR_ as priming and a supernatant of VVWR-infected cells as a boost; group II (five mice) received VVChimeric ENV as priming and C2V3C3 polypeptide as a boost; and group III (five mice) received VVChimeric ENV as priming and gp120AG as a boost. Mice were bled before immunization and 14 days after each immunization to determine binding and neutralizing antibodies.

### 4.5. Envelope-Specific Antibody Binding in Mice Sera

To analyse serum antibody responses in sera from immunized animals, Immuno MaxiSorp 96-well microplates (Nunc) were coated with soluble gp120AG (2 µg/mL), M.CON.S D11 gp120 (2 µg/mL), C2V3C3 polypeptide (2 µg/mL) or supernatant from VV_WR_-infected cells. After overnight incubation at 4 °C, the microplates were blocked with 2% gelatin (Bio-Rad). Mouse sera from days 0, 15, 30, 45 and 60 were heat-inactivated at 56 °C for 1 h, added to the microplates (1:200 final dilution in a total volume of 100 µL) and incubated for 2 h at room temperature. Goat anti-mouse IgG alkaline phosphatase antibody (Sigma) at 1:2000 dilution was added to the microplates as a secondary antibody. The colourimetric reaction was developed as described above. Negative controls were pre-immune serum and serum from mice immunised with VV_WR_. Positive controls were sera from individuals with HIV-1 and HIV-2. In this case, the secondary antibody was a goat anti-human IgG alkaline phosphatase antibody (Sigma). Sera with an optical density (OD) greater than that of the preimmune controls were considered positive.

### 4.6. Neutralization Assays

The neutralizing activity of mouse post-immunization sera was analysed in a one-round viral infectivity assay using TZM-bl against laboratory-adapted HIV-1SG3.1 (a Tier 1 subtype B isolate), two Tier 2 Env-pseudotyped HIV-1 isolates (TRO.11, subtype B, and CNE55, CRF01_AE) and a primary isolate of HIV-2 (03PTHCC6, group A). Env-pseudotyped viruses were generated by the transfection of Env-expressing plasmids into 293T cells using pSG3.1Δenv as a backbone and titrated in TZM-bl cells.

The neutralizing activity of mouse sera was tested in a one-round viral infectivity assay using a luciferase reporter gene assay in TZM-bl cells, as previously described [22]. Briefly, 10,000 cells in 100 µL DMEM supplemented with 10% heat-inactivated foetal bovine serum were added to each well of 96-well flat-bottom culture plates (Nunc) and allowed to attach overnight. Subsequently, 100 µL of each virus (200 TCID50/well) was incubated with heat-inactivated (56 °C, 1 h) mouse sera for 1 h at 37 °C in a total volume of 200 µL of growth medium (DMEM supplemented with 10% heat-inactivated foetal bovine serum) containing DEAE-dextran (19.7 µg/mL) and added to the cells. After 48 h, the medium was removed from each well and the plates were analysed for luciferase activity on a luminometer (TECAN) using the Pierce™ Firefly Luciferase Glow Assay Kit (ThermoFisher scientific). Wells with medium were used as background controls, and virus-only controls were included as infection controls. The effect of preimmune serum on infection was used as the baseline neutralizing activity. Percent neutralization was determined by calculating the difference in mean RLU (relative light units) between test wells containing post-immune sera and test wells containing pre-immune sera, after the normalization of the results using the mean RLU of the cell controls. The results were considered valid if the mean RLU of the virus control wells was ≥10 times the mean RLU of the cell control wells. To monitor the specificity of neutralization, each serum sample was also tested against pseudoviruses carrying the vesicular stomatitis virus (VSV) envelope. Neutralization assays were performed with sera collected after the third and fourth boosts. All mouse sera were tested for neutralizing antibodies once in triplicate at a dilution of 1:40. Neutralization assays were repeated for animals that neutralized at least one tier 2 isolate >30% in the first assay.

### 4.7. Epitope Mapping

Epitope specificities were assessed using a panel of HIV-1 TRO.11 pseudoviruses in which the major neutralizing epitopes in V2, the CD4 binding site and V3 were disrupted by mutagenesis at specific amino acids (N160A in V2, N278A in C3 and N332A in the V3 base) [48,70,71]. Mutant Env-pseudotyped viruses were produced by the transfection of Env-expressing plasmids into 293T cells using pSG3.1Δenv as a backbone and titrated in TZM-bl cells as described [72]. Target epitopes were identified by the reduction in neutralization scores compared to the wild-type pseudovirus.

## 5. Conclusions

In conclusion, a chimeric gp120 containing the C2V3C3 region of HIV-2 ALI and the remaining parts of an HIV-1 clade AE isolate elicited neutralising antibodies targeting some of the most broadly reactive epitopes in HIV-1 and HIV-2. Our results provide proof of concept for chimeric HIV-1/HIV-2 envelope glycoproteins as HIV vaccine immunogens.

## Figures and Tables

**Figure 1 ijms-24-09077-f001:**
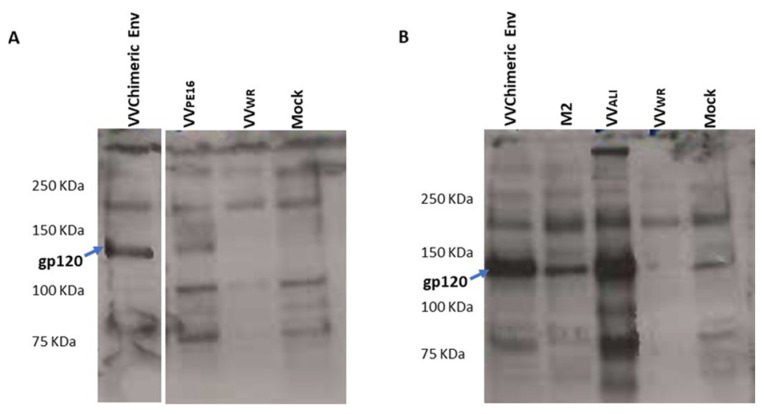
Reactivity of the HIV1/HIV2 chimeric gp120 expressed by recombinant vaccinia virus in Western blot against sera from individuals with HIV-1 (**A**) and HIV-2 (**B**). VV_WR_ is the wildtype vaccinia virus; VV_PE16_ is a recombinant vaccinia virus expressing the HIV-1 envelope and was used as HIV-1 positive control; VVALI (+) is a recombinant vaccinia virus expressing the HIV-2ALI envelope and was used as an HIV-2 positive control. VVChimeric ENV is the recombinant vaccinia virus expressing the chimeric gp120. M2 is a recombinant vaccinia virus expressing a soluble truncated envelope gp125 glycoprotein from HIV-2ALI and was previously described in Marcelino et al. [47]. Mocks are uninfected Hela cells. The molecular weights of the molecular weight marker bands (Precision Plus Protein Standards, Bio-Rad, Hercules, CA, USA) are given.

**Figure 2 ijms-24-09077-f002:**
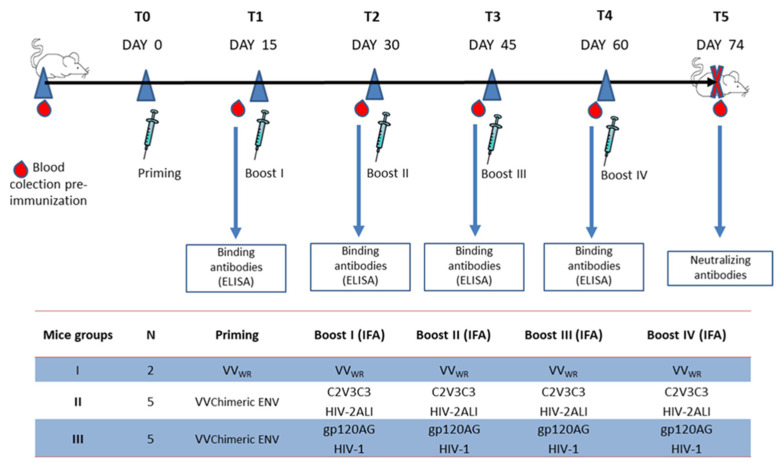
Schedule and regimens for immunization of mice and analysis of immune response. VV_WR_ is the wildtype vaccinia virus strain WR. VVChimeric ENV is the recombinant vaccinia virus expressing the chimeric envelope. C2V3C3 HIV-2ALI is the polypeptide comprising the C2, V3 and C3 regions of HIV-2ALI. gp120AG HIV-1 is the purified surface glycoprotein of a CRF01_AG isolate. The production and purification of these proteins were previously described by Calado et al. [22]. IFA, incomplete Freud adjuvant.

**Figure 3 ijms-24-09077-f003:**
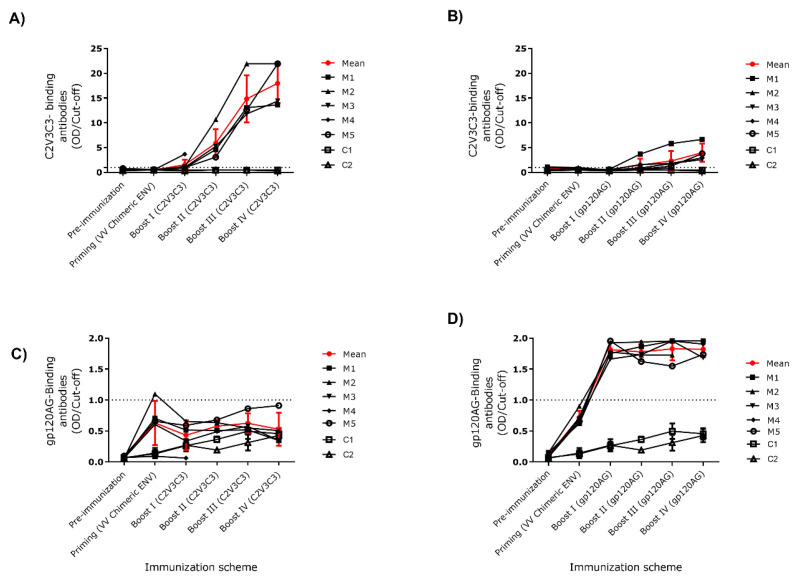
Kinetics of IgG binding response in mice primed with vaccinia virus expressing the chimeric Env and boosted with purified C2V3C3-polypeptide (**A**,**C**) or boosted with gp120AG (**B**,**D**). The binding antibody response was tested by ELISA assay against the C2V3C3 polypeptide (**A**,**B**) and against a soluble gp120 from a CRF01_AG isolate (gp120AG) (**C**,**D**). Control animals (C1 and C2) were primed with wild-type vaccinia virus (strain WR, VV_WR_) and boosted with supernatant of VV_WR_ infected cells.

**Figure 4 ijms-24-09077-f004:**
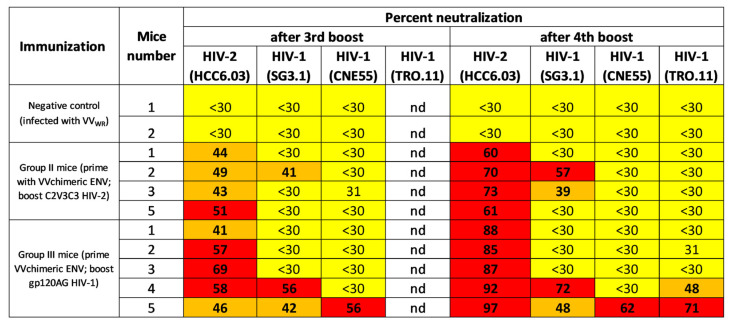
Neutralisation potency of mouse sera against tier 1 and tier 2 HIV isolates as assessed by the percentage of neutralisation at a 1:40 serum dilution. Animals were primed with vaccinia virus expressing the chimeric Env glycoprotein and boosted with either a C2V3C3 polypeptide from HIV-2 (group II) or a CRF01_AG gp120 (group III). Neutralisation results are shown after the third and fourth booster vaccinations. Yellow highlighting indicate less than 30% of neutralization; brown highlighting indicate ≥31–49% neutralization; red highlighting indicate equal or more than 50% of neutralization.

**Figure 5 ijms-24-09077-f005:**
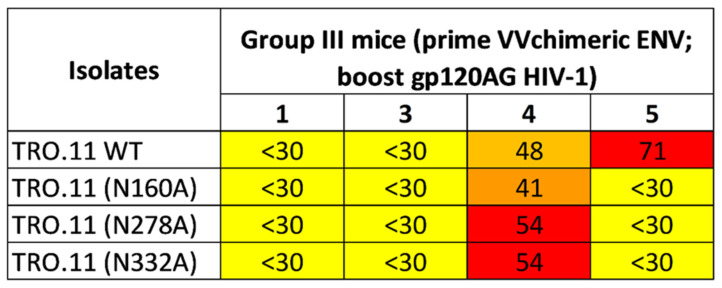
Neutralization potency of mice sera (% neutralization at 1:40 serum dilution) against HIV-1 TRO.11 mutant pseudoviruses. Animals were primed with vaccinia virus expressing the chimeric Env glycoprotein and boosted with a CRF01_AG gp120 (Group III). Neutralization results are shown after the fourth booster vaccination. Yellow highlighting indicate less than 30% of neutralization; brown highlighting indicate ≥31–49% neutralization; red highlighting indicate equal or more than 50% of neutralization.

**Figure 6 ijms-24-09077-f006:**
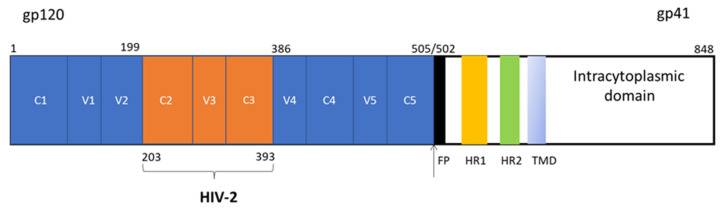
Schematic representation of the chimeric envelope glycoprotein expressed in vaccinia virus. Numbers indicate the position of the HIV-1 and HIV-2 regions in gp120. FP—fusion peptide; HR1—helical region 1; HR2—helical region 2; TMD—transmembrane domain.

**Table 1 ijms-24-09077-t001:** Binding of human monoclonal bNAbs to the chimeric Env gp120 and reference gp120 and gp140 glycoproteins in an ELISA assay.

HuMAbs	Epitopes	Chimeric Env gp120	M Con. S D-11 gp120	SF162 gp140
PG9	V1V2	0.58	**21.37**	0.23
PG16	V2	0.91	**1.66**	0.28
447-52D	V3	**1.52**	**23.05**	**16.05**
2G12	N-linked glycans	0.58	**23.05**	**16.05**
B12	CD4bs	**2.00**	**23.05**	**16.05**
VRCO1	CD4bs	0.36	**23.05**	**14.63**
VRCO3	CD4bs	0.38	**1.03**	**15.59**
HJ16	CD4bs	0.90	**23.05**	0.71
3BNC117	CD4bs	0.46	**23.05**	**12.54**
2F5	MPER (gp41)	0.91	0.42	**16.05**
HIV +	Multiple	**2.63**	**34.43**	**16.05**

Bold numbers indicate positive results as defined by OD/cut-off ≥ 1.

## Data Availability

Data collected for this study will be shared with publication upon reasonable request, with a signed data access agreement.

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
