# Peer review of "An HIV-1/HIV-2 Chimeric Envelope Glycoprotein Generates Binding and Neutralising Antibodies against HIV-1 and HIV-2 Isolates"

_ijms, 2023, doi:10.3390/ijms24109077_

Round 1
Reviewer 1 Report
An HIV-1/HIV-2 chimeric envelope glycoprotein generates binding and neutralising antibodies 2
against HIV-1 and HIV-2 isolates
The authors of manuscript intended to obtain bNAbs by immunizing Balb/c mice with recombinant vaccinia virus expressing chimeric HIV-1 and HIV-2 envelope envelope.
Methodologically, the study was well conducted. Quality od WB performed on infected HeLA-CCL2 lysates is not exceptional, but informative, indicating that vaccinia virus indeed carried the envelope sequence comprising both HIV-1 and HIV-2 domains.
Compared to the control envelopes M Con. S D-11 gp120 and SF162, the designed chimeric Env showed very limited binding of widely used monoclonal antibodies suggesting limited epitope exposure.
Following immunization and boosting regimens mice produced antibodies with specificities guided by used boosters showing overall immunogenicity of the chimeric Env. Two of five mice Mice boosted with HIV-1 gp120AG showed certain degree of cross neutralization for both HIV-1 and HIV-2.
Remarks:
The fact that chimeric Env binds B12 antibody is the only CD4 binding site antibody suggests the occluded-open conformation of the Envelope. Authors do not discuss this at all. Is the structure of the chimeric Env trimeric or monomeric?
It’s a pity that the authors have not used recombinant HIV-2 protein as an ELISA control, could the authors comment on the choice of the two controls used?
Could the authors discuss cross-neutralization of HIV-1 and HIV-2 by clinically tested bNAbs such as VRC01, 10-1074 or 3BNC117?
Author Response
The fact that chimeric Env binds B12 antibody is the only CD4 binding site antibody suggests the occluded-open conformation of the Envelope. Authors do not discuss this at all. Is the structure of the chimeric Env trimeric or monomeric?
The structure of the soluble chimeric gp120 that we have used in the ELISA assay should be monomeric as indicated by the molecular weight in the WB.
In the discussion we say: “The amino acids defining the B12 epitope are scattered in C1 (K97, L125), V2 (D167, D185, T202), C2 (N276), C3 (S365, P369), C4 (V430, T455, G458) and C5 (G473) [53-55]. Except for the substitutions E for P at position 369 in C3 and T for V at position 430 in C4, all amino acids are conserved in our chimeric construct, even those derived from HIV-2 in C2 (N276) and C3 (S365 and P369) (Figure S2). This high level of amino acid conservation in the context of a similar HIV-1 and HIV-2 gp120 structure [56] likely contributes to the binding of B12 to our chimeric gp120. While binding to B12 confirms that the chimeric gp120 exposes a CD4bs epitope, the lack of binding to the other antibodies that mimic the CD4 binding site suggests an altered CD4 binding site and impaired binding to the CD4 receptor in the cell surface.”
This paragraph acknowledges that the conformation of the soluble monomeric chimeric gp120 is likely different than the other monomeric envelope glycoproteins that bind most of the CD4bs antibodies. However, the envelope glycoproteins expressed by vaccinia virus in the mice cells in vivo should mostly be trimeric. Those are the envelope glycoproteins that are recognized by B cells and elicit the production of neutralizing antibodies.
It’s a pity that the authors have not used recombinant HIV-2 protein as an ELISA control, could the authors comment on the choice of the two controls used?
Recombinant HIV-2 envelope glycoproteins are not readily available. We used a recombinant HIV-2 envelope gp125 (M2) in the Western blot analysis with antisera from individuals with HIV-2. For the ELISA results shown in Table 1, we did not use the same M2 glycoprotein or any other HIV-2 envelope gp125 because all bNAbs tested were obtained from individuals with HIV-1 and previous studies have shown that these classes of bNAbs do not bind to HIV-2 envelope glycoproteins (see Decker et al. PMID: 16731954; our reference 60). The two monomeric HIV-1 envelope glycoproteins used as positive controls in the ELISA assay were obtained from the HIV Reagent Repository (NIH). Previous studies had shown that they bind to most of the bNAbs we intended to use in the binding assays, so they were considered to be good binding controls for comparison with our chimeric env. It should be noted that in general monomeric HIV-1 envelope glycoproteins fail to elicit broadly neutralising antibodies in animal models.
Could the authors discuss cross-neutralization of HIV-1 and HIV-2 by clinically tested bNAbs such as VRC01, 10-1074 or 3BNC117?
We agree that it would be interesting to explore the activity of these bNAbs against HIV-2. However, to our knowledge this has not been done. Julie Decker and coworkers (PMID: 16731954; our reference 60) have analysed four first generation CD4bs neutralizing antibodies (b12, F105, F91 and 15e) and found no activity against two isolates of HIV-2 (UC1, a group A virus, and 7312A, a A/B recombinant). They found however that some CD4 induced (CD4i) neutralizing antibodies, i.e., antibodies that target HIV coreceptor binding site and that are only exposed after CD4 binding, potently neutralize the same two HIV-2 isolates. We have not used any of these CD4i antibodies in this paper.
Reviewer 2 Report
In this study Taveira et al investigated the antigenicity and immunogenicity of a novel chimeric envelope gp120 comprising the C2, V3 and C3 regions of HIV-2 and the remaining parts of the HIV-1 envelope. This chimeric envelope was synthesized, expressed in vaccinia virus and injected to mice. The authors were able to elicit a neutralizing response against HIV-1 and HIV-2.
The study has novelty; however, significance is quite limited.
Major Comments:
1) In order to proof the concept, the neutralizing antibodies need to be tested against at least a few more highly prevalent HIV-1 strains.
2) The avidity of antibodies in neutralizing HIV-1 need to be compared with currently available one or two gp120-specific broadly neutralizing antibodies (bNabs), such as VRC01, VRC01-LS, 3BNC117, 3BNC117-LS or VRC07-523LS. This analysis will greatly enhance study significance.
3) The introduction and discussion about current available bNAbs, especially those against gp120 in the Introduction part will enhance the understanding of the subject.
Minor Comments:
1) English editing with a native English speaker will further enhance the flow of manuscript.

English editing with a native English speaker will further enhance the flow of the manuscript.
Author Response
Reviewer 2
In this study Taveira et al investigated the antigenicity and immunogenicity of a novel chimeric envelope gp120 comprising the C2, V3 and C3 regions of HIV-2 and the remaining parts of the HIV-1 envelope. This chimeric envelope was synthesized, expressed in vaccinia virus and injected to mice. The authors were able to elicit a neutralizing response against HIV-1 and HIV-2.
The study has novelty; however, significance is quite limited.
We thank the reviewer for the novelty part and reject the limited significance because a vaccine for HIV should be broad spectrum and include HIV-1 and HIV-2 as both viruses are lethal for infected and untreated individuals. There are very few vaccine studies against HIV-2. Our study confirms previous work showing that the C2, V3, and C3 envelope regions of HIV-2 contain the main neutralizing epitopes of HIV-2, and shows that these epitopes are adequately presented in the context of the HIV-1 envelope in a chimeric construct. These are significant results for the vaccine field. Regarding HIV-1, we agree that additional work is needed to fully demonstrate that our chimeric envelope may be part of an effective HIV-1 vaccine. An effective vaccine antigen should elicit a broad and potent Nab response in the majority of the tested animals which was not the case in our study. However, similar restricted results were also obtained in all other HIV vaccination studies performed so far with mice and other animals. Also, as mentioned below we would have liked to test the neutralizing response of the responding animals against many more HIV-1 strains of different subtypes. However, this was not possible because of the low amount of blood obtained from the animals. To circumvent this limitation, we have now produced seven monoclonal antibodies from the best responding mice and found that most of them neutralize HIV-1 and HIV-2 confirming and extending the original results presented in the paper. Please see below for details.
Major Comments:
1) In order to proof the concept, the neutralizing antibodies need to be tested against at least a few more highly prevalent HIV-1 strains.
We agree with the reviewer. However, this was not possible in our mice because they are small animals with a small amount of blood. We show all the binding and neutralisation experiments that we were able to do with the small amount of plasma that we were able to obtain from the animals. In order to be able to do more extensive research, as requested by the reviewer, we recently produced seven monoclonal antibodies (mAbs), all IgM, from the best responding mice. We tested the neutralising activity of these mAbs against five HIV-1 pseudoviruses belonging to highly prevalent subtypes (C, CRF07_BC and CRF01_AE) and four HIV-2 primary isolates from group A, which is the more prevalent group worldwide. The results were outstanding in that all but one mAb neutralised (>50%) at least three of the nine HIV isolates tested at 1 μg/ml (Table 1). One mAb neutralised seven isolates (77.8%) and three mAbs neutralised six isolates (66.6%). Collectively, these results confirm and extend the results presented in the paper and demonstrate that our chimeric antigen can generate potent and broadly neutralising antibodies that are effective against multiple isolates of HIV-1 and HIV-2. The neutralisation of different HIV-1 subtypes and HIV-2 by these mAbs suggests the existence of a common and conserved neutralising epitope in these viruses, most likely the CD4 binding site as discussed in the paper. Studies are underway to identify the neutralising epitopes of these mAbs. These results are shown only to the reviewer as the full characterisation of these mAbs is still ongoing and we are preparing a patent application to protect the intellectual property rights of these mAbs.
Table 1- Neutralizing ability of the seven monoclonal antibodies
Yellow shading indicates <30% neutralization, orange shading indicates ≥30%-<50% neutralization and red shading indicates ≥50% neutralization.
2) The avidity of antibodies in neutralizing HIV-1 need to be compared with currently available one or two gp120-specific broadly neutralizing antibodies (bNabs), such as VRC01, VRC01-LS, 3BNC117, 3BNC117-LS or VRC07-523LS. This analysis will greatly enhance study significance.
We agree and thank the reviewer for the suggestion. However, to be meaningful this type of comparative study needs to be done with mAbs and not with polyclonal antibodies. We plan to do this type of work with the newly generated mAbs (see above).
3) The introduction and discussion about current available bNAbs, especially those against gp120 in the Introduction part will enhance the understanding of the subject.
As suggested, the following text has been added to the Introduction: “The most common, potent and broad bNabs target the CD4bs, the V2 apex, and the V3-glycan patch epitopes in the HIV-1 envelope. CD4bs antibodies bind the CD4 binding site of gp120 and belong to two major families: those that mainly rely on the heavy chain complementary determining region 2 (HCDR2) to interact with HIV-1 gp120 (CD4 mimic bNAbs, e.g. B12, VRC01 class and 8ANC131/CH235 class) and those that instead use HCDR3 to contact gp120 (HCDR3-binding bNAbs, e.g. CH103) (reviewed in [23] and [36]). The epitope of V3-glycan targeting bNAbs is located at the base of the V3 region between N-linked glycans at positions 301 and 332. Their long HCDR3 loops are needed to reach the peptide backbone beyond the dense glycan patch defined by the GDIR sequence (amino acids 324-327). Examples of these bNAbs are PGT121, PGT128 and 447-52D. bNAbs targeting the V2 apex typically have extra-long HCD3 loops and sulfated tyrosine motifs required to contact the peptide epitope, a lysine-rich strand around positions 168–171, under the N-linked glycan at position 160 [23, 36]. Examples of such bNAbs are PG9, PG16 and VRC26.25.”
Minor Comments:
1) English editing with a native English speaker will further enhance the flow of manuscript.
The manuscript was submitted to the DeepLWrite site (https://www.deepl.com/write) to improve the quality of the English, and then read by a native English speaker.

Round 2
Reviewer 2 Report
Much better now